# Dehydroepiandrosterone and Its Metabolite 5-Androstenediol: New Therapeutic Targets and Possibilities for Clinical Application

**DOI:** 10.3390/ph17091186

**Published:** 2024-09-09

**Authors:** Tatiana A. Fedotcheva, Maria E. Uspenskaya, Darya N. Ulchenko, Nikolay L. Shimanovsky

**Affiliations:** Laboratory of Molecular Pharmacology, Pirogov Russian National Research Medical University, 1 Ostrovityanova St., Moscow 117997, Russia; m.uspenskaja@mail.ru (M.E.U.); motci@list.ru (D.N.U.); shimannn@yandex.ru (N.L.S.)

**Keywords:** androstenes, DHEA, 5-AED, nuclear receptors, membrane receptors, constitutive androstane receptor CAR, intracrine regulation, immunostimulation, regeneration, osteoporosis, prasterone

## Abstract

Dehydroepiandrosterone and its sulfate are the most abundant steroids in humans. The metabolism of dehydroepiandrosterone can differ significantly depending on the organ or tissue and the subtype of steroid receptors expressed in it. For dehydroepiandrosterone, as a precursor of all steroid hormones, intracrine hormonal activity is inherent. This unique feature could be beneficial for the medicinal application, especially for the local treatment of various pathologies. At present, the clinical use of dehydroepiandrosterone is limited by its Intrarosa^®^ (Quebec city, QC, Canada) prasterone) 6.5 mg vaginal suppositories for the treatment of vaginal atrophy and dyspareunia, while the dehydroepiandrosterone synthetic derivatives Triplex, BNN 27, and Fluasterone have the investigational status for the treatment of various diseases. Here, we discuss the molecular targets of dehydroepiandrosterone, which open future prospects to expand its indications for use. Dehydroepiandrosterone, as an oral drug, is surmised to have promise in the treatment of osteoporosis, cachexia, and sarcopenia, as does 10% unguent for skin and muscle regeneration. Also, 5-androstenediol, a metabolite of dehydroepiandrosterone, is a promising candidate for the treatment of acute radiation syndrome and as an immunostimulating agent during radiopharmaceutical therapy. The design and synthesis of new 5-androstenediol derivatives with increased bioavailability may lead to the appearance of highly effective cytoprotectors on the pharmaceutical market. The argumentations for new clinical applications of these steroids and novel insights into their mechanisms of action are discussed.

## 1. Introduction

Dehydroepiandrosterone (DHEA) and its sulfate (DHEA-S) are the most abundant steroid hormones in the human body. Of all the steroids, DHEA and DHEA-S circulate in the peripheral blood at the highest concentration, which is up to 10 µM [1]. DHEA and its metabolite 5-androstenediole (5-AED) have been known for a relatively long time as reservoirs of sex steroid hormones, but their own biological activity, as well as their specific targets, have not been precisely identified. Despite a surge of interest in the mechanism of their action in the 1990s of the last century and in the last decade, which revealed the unique properties of these androstene steroids for preventing sepsis in experiments [2], protecting from radiation [3], inhibiting tumor cell proliferation [4,5], and stimulating the viability of normal cells [6,7,8], there is no information about their specific mechanism of action.

DHEA serves as a precursor for the steroids androstenediol, androstenedione, testosterone (T), dihydrotestosterone (DHT), estrone, and 17β-estradiol (E2), and the biological effects of DHEA can partially intersect with the androgenic and estrogenic activities of these steroids [9,10]. The effect of DHEA is gender-specific [11], and this fact has been the subject of further clinical trials. Some studies indicate that daily long-term (for 12 months) supplementation of 50 mg of micronized DHEA increases the circulating levels of DHEA, DHEA-S, and androstenedione in both sexes [12], but testosterone levels increased to a lower limit of normal levels only in women [13]. Since the steroids 5-AED and DHEA are considered immunomodulators and radioprotectors, their effect on the host immune system is obvious. With age, the concentrations of these hormones decrease sharply, though not as sharply as those of testosterone and estradiol, which affects the overall immune resistance of an aging person. But the need for hormone replacement therapy using androstenes is not obvious, since rather little information has accumulated about their contraindications and particularly the risk of the development of testosterone- or estrogen-dependent tumor progression.

More intensive attention is desired for these hormones in the context of their role in the immune response, aging, and resistance to infection. They cannot be considered only as buffers or precursors; their specific targets should be identified. DHEA itself is an active hormone and can operate directly. The cytoprotective and anti-atherogenesis effect of DHEA on endothelial cells is estrogen receptor-independent [14]. There is evidence showing that the binding of DHEA to plasma membranes is highly specific, and the membrane receptors of sex steroids can play a critical role in DHEA and 5-AED nongenomic action.

The precise targets of DHEA and 5-AED have not yet been identified. According to the Drugbank database and our own opinion, these could be nuclear receptors—the AR androgen receptor (AR), estrogen receptor (ER), progesterone receptors (PRs), liver X receptor (LXR), peroxisome proliferator-activated receptor (PPAR), pregnane X receptor (PXR), or constitutive androstane receptor (CAR)—as well as the plasma membrane receptors, namely mARs, mERs, mPRs, GPERs, and GABA and NMDA receptors. It is generally accepted that DHEA is metabolized into androgens and estrogens in peripheral organs and mostly realizes its local androgen receptor-mediated action [15]. DHEA exhibits a binding affinity for ARs and ERs at low micromolar concentrations, but its affinity for ARs is from two- to threefold lower than that for ERβ [16]. One of the well-established pharmacological effects of DHEA is the antiglucocorticoid action [17], protecting mice against dexamethasone-induced atrophy of the spleen and the thymus [18]. The exact mechanism of the anti-glucocorticoid effect of DHEA is not known. DHEA does not bind to nuclear glucocorticoid receptors [18]. Downregulation of the mRNA of glucocorticoid receptors by DHEA may be the reason for the observed anti-glucocorticoid effects of DHEA [19].

DHEA regulates the inflammatory and cytokine responses to stimulation in various cell types and tissues and has immunomodulatory effects, which may have beneficial results in the treatment of autoimmune diseases [20]. DHEA is an uncompetitive inhibitor of G6PDH (Ki = 17 μM; IC50 = 18.7 μM) and lowers NADPH production, and accordingly, it is an inhibitor of NOX-dependent ROS production in various cell types [18]. In addition, DHEA induces expression of the PGC-1α and GLUT4 genes [21] and the CAR gene [22]. The activation of these genes may be useful for the treatment of such pathologies as diabetes, obesity, metabolic disorders, and insulin resistance. The stimulation of mitochondrial biogenesis is highly promising in the treatment of musculoskeletal failure.

Prospects for the clinical use of DHEA are associated with its local effect on the skin (regeneration and reduction of age-induced atrophy), namely the systemic effect of HRT on osteoporosis, reduction of menopausal symptoms, improvement of cognitive functions, as well as with its radioprotective and immunostimulatory activities. Until now, the clinical use of DHEA has been limited to prasterone, and 5-AED is not applied in clinics at all, but there are reasonable prospects for its implementation as an immunostimulant and radioprotector if it were possible to increase its bioavailability for oral administration through innovative technologies like nanoporous gels, nanoparticles, biopolymers, and chitosan constructions. This review examines the key mechanisms of the action of DHEA and 5-AED and outlines future prospects for their clinical use.

## 2. Biosynthesis and Transformation of DHEA

The synthesis of DHEA and its sulfate DHEA-S occurs in the adrenal glands, and only a small part (about 8–10%) is produced by the gonads [23]. DHEA and DHEA-S are synthesized de novo in the brain and spinal cord, since their concentrations in these organs are higher than in plasma and remain high after adrenalectomy and gonadectomy [24].

DHEA is produced during steroidogenesis mainly by two classes of steroidogenic enzymes: P450s and hydroxysteroid dehydrogenases [25]. Additionally, DHEA can be sulfated to DHEA-S by the enzyme steroid sulfotransferase (SULT), while the conversion of DHEA-S back to DHEA is mediated by steroid sulfatases (STSs) (Figure 1a). The activity and expression of hydroxysteroid dehydrogenases in the prostate determines whether the final effect is androgenic or estrogenic [26]. Most of DHEA is sulfated, and part of DHEA is converted into androstenediol by AKR1C3 (Figure 1b). Additional DHEA and androstenediol can both be converted into their corresponding Δ4 products by the hydroxysteroid dehydrogenases isoform 3B2, yielding 4-androstenedione (A4) and testosterone, respectively. A4 is an additional substrate for AKR1C3 to produce testosterone [27].

The level of DHEA-S in human blood is roughly 400 times higher than that of DHEA itself. DHEA-S is a reservoir of DHEA, although some studies indicate higher biological activity of the sulfate form. For example, DHEA-S was shown to inhibit G6PD more strongly than DHEA [28]. Further research is desirable to solve the question of why DHEA-S may be more active than DHEA.

In the early 1980s, Fernand Labrie, a renowned endocrinologist, noticed that the prostates of patients suffering from prostate cancer had high levels of DHT even after castration [9]. He demonstrated that the prostate contains hydroxysteroid dehydrogenases necessary for converting DHEA and DHEA-S into their androgenic and estrogenic metabolites, making them capable of producing definite local effects [29]. With intracrine regulation, DHEA formed in the adrenal glands, when entering the blood of general circulation, reaches the peripheral tissues where, inside the cells themselves, with the help of appropriate enzyme systems, it is transformed into the biologically active steroid hormones estradiol or testosterone which, without leaving the cells, have a biological effect. In other words, they do not penetrate into the extracellular space or the general blood flow [23]. The physiological role of DHEA and DHEA-S has been established to consist of serving as precursors for peripheral synthesis of the sex steroids DHT, T, and E2. The levels of these precursors determine the DHT or T and E2 concentrations in the peripheral tissues, although they have almost no effect on the levels of these hormones in plasma [16].

From five to seven percent of DHEA in both men and women is converted into androstenes, particularly Δ5-androstenediol (5-AED). The formation of this steroid is of great biological significance since it binds to both androgen and estrogen receptors, and 5-AED can be called a “hermaphrodiol” due to its high affinity for both of these receptors [23]. Unlike DHEA, 5-AED is not used as a drug, and no completed clinical trials have been carried out, but DHEA has undergone 59 clinical trials. Most studies are associated with its use in pathological conditions, such as vulvovaginal atrophy associated with menopause and breast cancer treatment in elderly women.

The differences in the effects of DHEA on cells may be due to the fact that steroids may be metabolized differently in various cell types. Thus, HeLa cervical cancer cells show heterogeneity in metabolizing radiolabeled testosterone with the alternative 17-oxosteroid and Δ4 pathways, metabolic pathways which differ from those of prostate cancer cells [30]. Also, men and women have different metabolisms. Thus, during pregnancy, estriol, which has a 16α-hydroxy group, is the predominant metabolite of DHEA [25].

Not only DHEA itself but even its metabolites primarily have the final effect, which confirms the idea that the pharmacodynamics of DHEA and 5-AED depend on metabolizing enzymes, as well as ARs, ERs, GRs, and other nuclear, cytosolic, and plasma membrane receptors.

Recent studies have established the occurrence of cross signaling between PXRs, CARs, and RXRs [31]. These receptors are also the targets of DHEA, which indicates that DHEA and AED affect not one but several receptors at once, and the functional response depends on the presence in the tissue of one type of receptor or another [32].

The clinical efficacy and the side effects of DHEA may also vary because the appropriate targets may bind with DHEA and trigger biological responses differently depending on the tissue type. The targets, indications, and possible side effects of these hormones will be discussed below.

## 3. DHEA and 5-AED Targets

According to the Drugbank database, there are some established targets for DHEA (see Table 1), but there are none for 5-AED. The targets of DHEA are nuclear and plasma membrane receptors, enzymes, and possibly other cellular proteins.

### 3.1. Nuclear Receptors of Androstenes

One of the main DHEA targets is nuclear estrogen receptor β (ERβ). The next-strongest identified target of DHEA is the androgen receptor (AR). DHEA exhibits a binding affinity for AR, which is two-to-threefold lower than that for ERβ. DHEA binds to ERα with an equilibrium binding constant (Kis) of 1.1 μM and to ERβ with a Kis of 0.5 μM, with the preferential agonism for ERβ and an EC_50_ of approximately 200 nM [16].

Androstenediol binds to ERα with a Kis of 3.6 nM and to ERβ with a Kis of 0.9 nM [47]. The Kis values for the AR and ER isoforms are presented in Table 2.

As shown in Table 2, the constant of binding DHEA to ARs (~1.2 µM) is much higher than that of T (0.5 nM). The constants of binding DHEA to ERα (~1.0 µM) and to ERβ (~0.5 µM) are much less than that of T (>5 µM). As for the other nuclear steroid receptors, with the tested concentrations being as high as 5 μM, there was not even minimal detectable binding of DHEA to either the glucocorticoid receptor or progesterone receptor [16].

In the same assay, the binding affinity of 5-AED was tighter for the ERs than that of DHEA, being 3.6 nM and 0.9 nM for ERα and ERβ (for 5-AED) and 245 nM and 163 nM for DHEA, respectively [16,39]. DHEA also binds with PXRs and SXRs with a Kis of 50–100 µM and PPARα in a micromolar range [39,41]. These studies confirm that 5-AED is more estrogenic than DHEA when comparing their relative binding affinities (RBAs) to ERs. In an AR transactivation assay but not in a binding assay, 5-AED was much less effective than DHT (EC50 of 2969 nM and 0.06 nM, respectively). On the contrary, 5-AED activated estrogen receptors at a lower concentration, with an EC50 of 2.5 and 1.7 nM for ERα and ERβ, while the same constants for DHT were 112 and 409 nM, respectively [48]. In general, 5-AED bound to and transactivates estrogen and androgen receptors with the same general rank order of potency as DHEA [48].

In spite of high binding affinities for ARs and ERs, DHEA strongly inhibits the proliferation of the cervical cancer cells HeLa, but its effect is not mediated by androgen or estrogen receptor pathways, since the antiproliferative effect is not abrogated by the inhibitors of these receptors [50]. This indicates that the pharmacological activity of DHEA is realized through other nuclear receptors.

DHEA and DHEA-S also regulate a number of hepatic NRs, namely PPARα, CARs, and PXRs, which regulate the transcription of CYP genes and other foreign compound-metabolizing enzymes [39].

The positive anti-diabetic action of DHEA is also brought about through PPARγ and its transcriptional coactivator PGC-1α. DHEA activates the AMPK-PGC-1α-NRF-1 and IRS1-AKT-GLUT2 signaling pathways and therefore prevents glycolipid metabolic disorder [51].

The micromolar effective concentrations of DHEA for 50% activation of PPARα (PPARα regulates the peroxisomal beta-oxidation of fatty acids), CARs, and PXRs are higher than those for ERs but achievable, since circulating DHEA-S is merely in the micromolar range. Prough, R.A. et al. suggested that DHEA and DHEA-S act as direct ligands for hepatic NRs and G protein-coupled receptors [39].

It is necessary to identify more specific receptors for DHEA and 5-AED and determine the Kis, which will allow one to create and synthesize more specific ligand targets for them. Due to the extremely low toxicity of these steroids and their intracrine action, future studies should be aimed firstly at development of the dosage forms for local external use in the case of DHEA and secondly at development of the dosage forms for the oral uptake of 5-AED. Both directions have good prospects.

### 3.2. Plasma Membrane Receptors of Androstenes

The pleotropic action of DHEA can be realized through different plasma membrane receptors, but the binding capacities and subsequent signaling are thus far poorly understood [34,52].

DHEA and DHEA-S may directly bind to DHEA-specific GPCRs in endothelial cells and to various neuroreceptors, such as γ-aminobutyric-acid-type A (GABA(A)), N-methyl-d-aspartate (NMDA), and sigma-1 (S1) receptors (NMDAR and SIG-1R). DHEA G protein-coupled receptors in endothelial cell plasma membranes consequentially regulate eNOS activity through Galpha(i2) and Galpha(i3) [53].

DHEA directly binds to membrane neurotrophin receptors. Saturation experiments and Scatchard analysis of radiolabeled DHEA binding to membranes isolated from HEK293 cells transfected with the cDNAs of TrkA and p75NTR receptors showed that DHEA binds to both membranes with a Kd of 7.4 nM and 5.6 nM for TrkA and p75NTR, respectively. The binding of DHEA to NGF receptors mediates its anti-apoptotic effects in the form of blocking TrkA expression through the RNAi reversed cytoprotective action of DHEA [54].

DHEA and 5-AED, like many other steroid hormones, could possibly bind to not only nuclear but also membrane ARs, ERs, and membrane progesterone receptors (mPRs) and realize their rapid nongenomic action. Future in silico, in vitro, and in vivo studies are required to determine these possibilities (e.g., binding capacities for membrane ARs (mARs)), whose crystal structures have not yet been determined, membrane ERs (G protein-coupled estrogen receptor 1 (GPER1)), and progesterone membrane receptors (Progesterone receptor membrane component 1 and 2 (PGRMC1 and PGRMC2) and mPR αβγδε) [55]. Special attention should be paid to the interaction of DHEA and 5-AED with progesterone membrane receptors and possibly progesterone nuclear receptors, as the structure similarity of their potential ligands is high enough (similarity between DHEA and some progestins).

## 4. DHEA and 5-AED in ATC Classification

Whereas the DHEA metabolite 5-AED is not used in clinical practice due to its instability, DHEA in the form of the vaginal gel prasterone is included in ATC classification; it belongs to anabolic steroids, namely androstane derivatives, along with such well-known anabolic steroids as norethandrolone, methenolone, androstanolone, and stanozolol [56]. In the USA and Canada, the drug is classified as belonging not only to the G03XX01 group but also the G03EA03 group “Androgens and estrogens” and the androstene derivatives A14AA (“androstan derivatives”) group of anabolic steroids.

According to the WHO classification, prasterone belongs to the L02AX group (L: Antineoplastic and Immunomodulating Agents; L02: Endocrine Therapy; L02A: Hormones and Related Agents; L02AX: Other Hormones) [57].

Thus, at the same time, prasterone belongs to 4 ATC groups: A14AA07 prasterone, G03XX01 prasterone, G03EA03 prasterone and estrogen, and L02AX antineoplastic and immunomodulating agents.

Based on the structure analysis of DHEA and 5-AED and the belonging of DHEA to the ATC, they may have possible implications in the treatment of infectious and other diseases such as osteoporosis, growth retardation, cachexia, and cancer. Further research is needed to develop doses and regimens of administration for these steroids as well as drug formulations suitable for each disease or pathology.

## 5. Current Clinical Use of DHEA and 5-AED

Recently, DHEA came into use as a drug in Western Europe. In the USA and Canada, it has been used since 1997. DHEA-S was marketed as an injectable form in Italy in the 1950s under the name Astenile [58].

Many clinical studies with DHEA as a drug were carried out, and in most of them, a dose of 50 mg/day of DHEA was used. Based on its daily production in the body, this dose is considered optimal [12]. A dose of 300 mg/day is the maximum dose. There are published results for clinical studies on the application of vaginal gel and the oral form of DHEA.

### 5.1. DHEA Vaginal Gel

The efficacy of DHEA (Intrarosa) in moderate-to-severe dyspareunia, a menopausal symptom of vulvar and vaginal atrophy, was examined in two primary 12 week placebo-controlled efficacy trials. Their main results were published [59,60,61,62,63,64,65], and the summary of these results is given in Table 3.

All of the differences between the baseline and endpoints were statistically significant and indicated improvement in the symptoms of vulvar and vaginal atrophy after treatment with 6.5 mg of Intrarosa.

Clinical study № NCT 01376349 on breast cancer and gynecological cancer patients with DHEA as supportive care showed positive results. Treatment with DHEA (6.5 mg vaginal gel) for 12 weeks in 118 patients decreased the severity of vaginal symptoms. In this study, the primary outcome was the severity of the dryness or dyspareunia. The response to DHEA administration was positive, being the same for low (3.25 mg) and high (6.5 mg) doses.

### 5.2. Oral DHEA

Clinical studies with the oral form of DHEA also demonstrated its positive effect on the hormonal profile, menopausal clinical symptoms, sexual dysfunctions, and bone and skeletal mass. The main action of the hormone was in restoring the decreased levels of endogenous DHEA and DHEA-S [65,66]. The doses were 50, 100, and 200 mg. All of these doses were tolerated well and did not cause any serious side effects. All of the clinical studies revealed the safety and positive effects of DHEA treatment on vaginal atrophy parameters, bone mass, and the endogenous DHEA level.

The oral administration of prasterone at doses of up to 1.6 g per day provoked no overdosage in postmenopausal women [58,67,68]. Thus, DHEA in four divided doses (400 mg doses) was administered to six postmenopausal women for 28 days in a double-blind placebo-controlled study. The serum androgen concentrations after the first dose of DHEA increased rapidly and reached a maximum at 180–240 min. As a result, the increase over the baseline for DHEA was sixfold (from 5.0 to 28.8 nmol/L), while it was 12 fold for DHEA-S, 14 fold for androstenedione (from 1.4 to 19.5 nmol/L), 2.5 fold for testosterone (0.7 to 2.2 nmol/L), and 13 fold for dihydrotestosterone (from 0.2 to 2.73 nmol/L). The concentrations of estrone, estradiol, and sex hormone-binding globulin (SHBF) were unchanged. The assessments at weekly intervals revealed a further increase in the concentration of all androgens, which was maximal at 2 weeks and remained markedly elevated for up to 4 weeks. However, during the 4 week experiment, the levels of LH and FSH, as well as the body weight and percentage of body fat, did not change.

The use of a standard dose (50 mg per day) of DHEA orally in a 52 week clinical study did not significantly alter the lipid profile or insulin sensitivity. The drug had no influence on the endometrium in postmenopausal women [69].

Another clinical trial revealed a positive effect of DHEA on the bone mineral density (BMD) in young women with anorexia nervosa. This study (N° NCT00310791) was conducted on 70 girls (aged 13–27 years, with the treatment group involving 35 girls and the placebo group having 35 girls) with 50 mg of oral micronized DHEA combined with conjugated equine estrogens (0.3 mg daily) every day for the first 3 months, followed by an oral contraceptive (20 μg ethinyl estradiol/0.1 mg levonorgestrel) for 9 months. Positive effects from the treatment with DHEA + estrogen/progestin were seen primarily in girls with a body mass index above 18 kg/m^2^. With treatment for 18 months, DHEA + combined oral contraceptive (COC) stabilized the BMD, distinct from a decrease found in the placebo group [70,71].

Several specific blinded, placebo-controlled randomized clinical studies with oral prasterone (200 mg per day) were also conducted with female patients with active systemic lupus erythematosus (SLE). The results showed a reduced risk of auto-immune flare, breast cancer, and death from any cause in the treated groups. Oral prasterone was shown to restore the serum DHEA-S levels in patients with SLE. Women with SLE thus often have reduced levels of this prehormone [72] (number NR 98-2-301). DHEA restored the decreased endogenous DHEA level and improved the outcome of the disease. Although extensive clinical trials have not yet been performed, there is more and more evidence that DHEA is effective for the treatment of patients with poor ovarian reserve (POR) and may be used before ovarian stimulation in in vitro fertilization (IVF) protocols to increase the number of successful outcomes [1,2,3]. In the case of ovarian insufficiency, DHEA supplements are usually prescribed orally at a dosage of 50–90 mg per day for 1–3 months prior to the subsequent IVF cycle. DHEA administration not only helps to increase the success rate of IVF but can also increase the number of retrieved oocytes, fertilized oocytes, and top-quality embryos [2,3] and can improve such parameters as the endometrial thickness or oocyte quality [4]. These clinical findings may allow one to consider DHEA as an effective tool in the management of infertility caused by POR.

Currently, 5-AED is not used in clinical practice; clinical trials of a 5-AED-based drug were terminated. Androstenediol was tested for use as a radiation countermeasure by Hollis-Eden Pharmaceuticals under the proposed brand name Neumune in the treatment of acute radiation syndrome. The main mechanism of action was the stimulation of innate immunity and alleviation of neutropenia and thrombocytopenia [73].

Some DHEA-based substances have passed clinical trials. Table 4 presents the main derivatives and their dosage forms under code names which are undergoing preclinical and clinical trials.

The table shows that the most impressive clinical trials were carried out with Triolex for the treatment of collagen-induced arthritis, Alzheimer’s disease (Phase 3, 2024), and Parkinson’s disease (Phase 1 and 2, 2024).

In the USA, the experimental drug Neumune, which has a chemical structure of 5-androstenediol (5-AED), was widely studied as a radioprotector and radiomitigator. However, the trials did not reach the FDA approval phase. In laboratory animals of various species, including monkeys, 5-AED demonstrated a pronounced radioprotective effect under conditions of acute and long-term irradiation, as well as under combined exposure to radiation and chemical and biological damaging factors [73,86]. The drug exhibited anti-radiation properties when injected both before (48–24 h) irradiation and in the early (1–4 h) periods after radiation exposure. It is well tolerated and effective when administered subcutaneously and orally, and the side effects are minimal. Neumune was allowed to be tested on humans. The first stage of clinical trials of Neumune™ (double-blind randomized trial) was performed on 129 healthy volunteers of different ages and yielded highly promising results. Injections of Neumune™ showed excellent tolerance and noticeably increased the number of neutrophils and platelets circulating in the peripheral blood. The side effects were negligible, with only a minor local reaction at the injection site [73].

Fluasterone was mentioned for the last time in Pubmed in 2010 in a pharmacokinetics study on dogs, which demonstrated some advantages of the subcutaneous route of administration to peroral and the intravenous route due to prolonged release and increased retention through 24 h [18]. The compound was tested for use in the treatment of psoriasis and psoriatic disorders, hyperlipidemia, metabolic disease, cancer and tumors (unspecified), and obesity. Recently, Fluasterone has been approved by the FDA for a clinical study on the efficacy and safety of buccal tablets in the control of hyperglycemia in adults with Cushing syndrome [87].

Triolex (HE3286) has passed four different clinical trials of phases 1 and 2. Potential mechanisms of action for HE3286 could be the regulation of NF-kB or an increase in the production of regulatory T cells (Treg cells). Interestingly, Triolex has a similarity score (based on the molecule chemical structure) of 0.7 with DHEA and a similarity score of 0.73 with FDA-approved progestin, used for the contraceptives Lynestrenol and Gestodene [88,89]. Thus, the anti-inflammatory properties of Triolex could be due to binding with progesterone receptors.

The neurosteroid BNN27 is a novel synthetic derivative of DHEA, a synthetic C-17-spiro-dehydroepiandrosterone analogue. The compound is being studied and has recently been shown to have anxiolytic activity (2024) [79]. BNN27 is considered to lack any androgenic and estrogenic effects since it does not activate the appropriate steroid hormone receptors [80]. BNN27 is able to cross the blood–brain barrier [80]. Compared with DHEA, BNN27 exhibits a higher affinity for the TrkA and p75NTR receptors of the nerve growth factor (NGF) but does not affect the severity of pain [78]. BNN27 is being extensively studied for intranasal administration in different dosage forms. It was shown that chitosan-coated nanoemulsions can deliver more BNN27 to the brain than chitosan-coated liposomes [90].

As shown in Table 4, the most extensive actual studies were carried out with BNN27 and Triolex. Many investigations have been performed with DHEA itself. In most of them, a dose of DHEA of 50 mg/day was used, based on its daily production in the body, and this dose is considered optimal, while a dose of 300 mg/day is the maximum [12].

As for further clinical trials, it is extremely important to consider the route of administration, since the bioavailability of DHEA is highly dependent on the way the drug is delivered. Oral uptake seems to be more effective, as in a clinical study on premenopausal women, the levels of DHEA in the blood were compared with the vaginal route versus oral micronized or unmicronized DHEA (150 mg). With vaginal administration, the levels of circulating DHEA, DHEA-S, and testosterone in the blood did not change, and with oral administration, the concentrations of these three steroids increased [91].

The bioavailability and tissue distribution of DHEA and 5-AED after oral and external use should be determined in future studies. At present, it is known that the bioavailability of DHEA in cynomolgus monkeys after the administration of a single oral dose of 50 mg is 3.1% [92]. The bioavailability of 5-AED has not been widely studied, although there is evidence that it is low. The methods for its determination in the blood have thus far been developed only at the experimental laboratory level [93].

Based on the information about clinical trials, it can be summarized that local application (intravaginal or transdermal) increases the concentration of DHEA itself at the injection site without increasing the concentrations of circulating DHEA, DHEA-S, and testosterone in the blood. Therefore, topical use of 5–20% DHEA in the form of a gel or ointment is promising.

When DHEA is administered orally, several metabolites are formed. The major circulating metabolites of DHEA are DHEA-S, androstane-3 alpha,17-beta-diol-glucuronide, and androsterone glucuronide [92]. Steroids are metabolized quickly, and therefore, for oral use, DHEA esters or esters of 5-AED could be quite effective for a longer release and prolonged action of the active steroid substance.

## 6. Prospects and New Clinical Applications of DHEA and AED

After decades of oblivion, there has been a surge of interest in androstenes [23]. New information has evolved for such aspects as the action of androstenes as regenerative (anti-aging), neuroprotective, and radioprotective and stimulating the immune system [94]. Both DHEA itself and its synthetic derivatives may have a cytotoxic effect on tumor cells [20]. Up to now, in the scientific literature, there are limited data about the effect of DHEA or 5-AED on the proliferation of normal and cancerous cells, as well as the targets of their action. It is assumed that the targets for implementation of the cytotoxic effect may be nuclear and membrane receptors (ARs, ERs, and peroxisome proliferator-activated receptor (PPAR)) [22], as well as various enzymes like G6PD, NADP oxidase, and NOS [18]. The main clinically significant and promising pharmacological effects of DHEA will be discussed below.

Due to intracrine regulation, DHEA is able to exert a unique effect, namely increasing the level of androgens if it is reduced to normal and no more than is necessary. This is evidenced by the latest clinical data on patients with miscarriages. DHEA supplementation restored the serum estradiol level, which resulted in a statistically significant reduction in miscarriages [95]. Conversely, a clinical case was described in which DHEA supplementation restored the testosterone level, and this led to successful pregnancies when using oral DHEA treatment for hypoandrogenemia in a 30 year-old female suffering from five recurrent miscarriages [95]. These data again remind us that DHEA can bind and exert its action through progesterone receptors, since progestins are the first-line drugs used in the treatment of miscarriages.

### 6.1. Androstenes for Treatment of Mucosal Atrophy

Clinical trials confirmed the effectiveness of DHEA at a daily intravaginal administration of 0.50% (6.5 mg) for vulvovaginal atrophy and dyspareunia [96]. Moreover, the effect of DHEA does not extend to the endometrium; it works only in the vagina, and serum sex steroids remain within the biologically low postmenopausal reference range, which rules out any stimulation of the already atrophic endometrium [97]. The mechanisms of the positive action of DHEA on mucosal atrophy are still unknown.

In 2007, a clinical trial was initiated to elucidate the mechanisms of action of DHEA and its biotransformation into active steroids and steroid metabolites and to evaluate the positive effects of DHEA on the bioavailability of IGF-1 and the inhibition of IL-6 production [98]. However, either concrete results were not achieved, or they were not published. Later, in 2019, the same group of scientists from the USA analyzed four studies and concluded that, in general, DHEA hormone replacement therapy has sex-specific effects (i.e., it has gender specificity and is more favorable for women) [11]. Further research is needed to evaluate whether DHEA has a more favorable risk-benefit profile for women than estrogen therapy.

The regenerating effects of DHEA have also been demonstrated for other tissues in uncontrolled trials. Since prasterone has proven effectiveness and safety in the treatment of vaginal atrophy, great prospects may be associated with its use in mouth mucosa atrophy, atrophic rhinitis, and gastrointestinal or respiratory tract atrophy.

### 6.2. Skin Anti-Aging Effect of Androstenes

The regenerating effect of DHEA toward fibroblasts from different tissues is well known. The precise mechanism of this stimulating effect has not been identified, but some studies showed that the extended lifespan of human fibroblast Wi-38 cells which occurs when these cells are maintained in culture medium supplemented with the steroid hormone dexamethasone is accompanied by a suppression of p21 levels, which normally increase as these cells enter senescence [99]. The same mechanism of the regenerating effects (the delay of senescence) possibly operates in the case of DHEA. Also, anabolic action toward the skin has been shown [100].

More importantly, DHEA not only stimulates the proliferation and viability of fibroblasts [20] but also upregulates the levels of skin-regenerating proteins in cultured human skin fibroblasts. DHEA increases collagen production in a dose-dependent manner, with the maximal effect at 10 µM. It also increases the levels of alpha1 procollagen mRNA up to 1.6 fold.

DHEA decreases the expression of collagen-metabolizing enzymes. The levels of collagenase mRNA decreased in response to DHEA treatment by 40%, whereas those of stromelysin-1 mRNA increased up to 2.4 fold compared with its controls. The maximum activation of the stromelysin-1 encoding gene occurred at a DHEA concentration of 1 µM, 4.5 fold higher than that in the control.

The effects of DHEA on the expression of these genes occurred at micromolar concentrations, which were not much higher than those in human plasma 10^−8^–10^−6^M). This suggests appropriate benefits from local skin application of DHEA [101].

The in vivo skin-regenerating effect of DHEA was demonstrated in a Korean study by Dr. Shin M.H. et al. DHEA (5%) in ethanol:olive oil (1:2) was tested on 12 volunteers, being topically applied to their buttock skin three times per week over 4 weeks. There was a significant increase in the expression of procollagen α1(I) protein and mRNA in both aged and young skin. Topical DHEA significantly decreased the basal expression of MMP-1 protein and mRNA and increased the expression of TIMP-1 protein in aged skin [102].

### 6.3. The Effect of Androstenes on the Musculoskeletal System

DHEA and 5-AED have regenerating anabolic action toward muscle tissue and skin [103]. DHEA improves muscle flap microcirculation and hemodynamics [104]. The administration of DHEA has positive effects on muscle mass and strength, as well as physical performance parameters [105].

Numerous clinical studies and experimental studies on animals have confirmed the promise of DHEA for the treatment of osteoporosis [11,106,107,108,109]. It is noteworthy that there is great similarity in the chemical structure of prasterone (DHEA) and nandrolone (Retabolil). In turn, osteoporosis of various origins is the main indication for nandrolone.

Due to intracrine regulation (i.e., formation at the site of action of precisely those DHEA metabolites which are deficient due to pathology) and subsequent implementation of the corresponding effect (estrogenic, androgenic, anabolic, anti-inflammatory, and immunostimulating), DHEA has good prospects for the treatment of osteoporosis, one of the main causes of which is a decrease in the level of sex hormones.

Clinical studies demonstrated an increase in BMD after DHEA treatment, due to the ability of DHEA to increase osteoblast activity and the expression of insulin-like growth factor 1 (IGF-1) [110] (Figure 2). The improvements in BMD in response to DHEA are accompanied by the suppression of bone resorption and the stimulation of bone formation. The osteogenic effects in elderly people are consistent with the estrogenic and androgenic activities of DHEA, which serves as a precursor to active androgens and estrogens in local tissues such as bones. DHEA replacement may also increase the level of IGF-1, which may contribute to its anabolic effects [111].

In vitro experimental studies on primary cultures of human bone marrow-derived mesenchymal stem cells (hMSCs) indicate that DHEA may be beneficial for bone health, owing to its inhibitory effect on skeletal catabolic IL-6 and stimulation of osteoanabolic IGF-I-mediated mechanisms (Figure 2). DHEA stimulates osteoblastogenesis. The in vitro stimulation of both osteoblastogenesis and IGF-I gene expression by DHEA in hMSCs requires the presence of an IGF-I receptor and activation of the PI3K, p38 MAPK, or p42/44 MAPK signaling pathways. The inhibition of inflammatory and catabolic IL-6 synthesis in hMSCs by DHEA is more pronounced than that by estradiol or dihydrotestosterone [112].

Another mechanism of the regenerating action of DHEA on the bone mass is its ability to increase the expression of Runx2 and osterix, thereby elevating the expression of osteocalcin and collagen 1. DHEA also upregulates osteoblast differentiation via induction of the expression of osteoblastogenesis-related genes and Tregs [113] (Figure 2).

### 6.4. The Use of Androstenes as Antitumor Agents

The prospects for using DHEA and 5-AED in the adjuvant therapy of tumors are debatable since the data are contradictory. Steroids can have both cytoprotective and cytotoxic effects, depending on the tissue and the presence of steroid receptors. Recent studies have revealed the potential of DHEA as an anti-stemness antitumor agent which can enhance the cytotoxic effects of irinotecan [114].

Some clinical results demonstrate antioxidant and anti-DNA-damaging effects of DHEA in patients suffering from pancreatic cancer. The protective effect of DHEA against pancreatic cancer is due to the ability of DHEA to prohibit accumulation of the 8-hydroxy-2′-deoxyguanosine (8-OHdG) DNA adduct. This cytoprotective function of DHEA may be one of its physiological roles in an organism [115].

Other clinical studies showed that serum DHEA-S levels of 90 μg/dL or more (≥2.43 μmol/L) are a risk factor for the progression of IV breast cancer treated with third-generation aromatase inhibitors or adjuvant tamoxifen citrate [32]. In contrast to the stimulatory effect observed in ER-positive cells, the proliferation of ER-negative cells is inhibited by DHEA-S. The final cellular response will depend on the expression of receptors and the hormones available for binding, because DHEA-S also regulates proliferation not only through the ERs but also through the ARs.

There is rather little information about the effects of DHEA and 5-AED on the proliferation of different cell types. More often, the IC50 values are given for their derivatives. The cytotoxic effect of derivatives may even exceed the effect of classical cytostatics in vitro. The derivatives of A-homolactam D-homolactone androstane, synthesized from DHEA, were superior in terms of their IC50 on PC3 cancer cells versus doxorubicin and formestane [116].

As a direct DHEA metabolite, 5-AED has a more distinct therapeutic effect in the treatment of estrogen-dependent tumors, as it inhibits the estrogen-induced proliferation of hormone-dependent breast cancer cell lines through an androgen receptor-mediated mechanism [116,117,118]. Since 5-AED binds with a higher affinity to the ERs and ARs than DHEA, a nuclear receptor-mediated response occurs.

It was previously shown that the presence of androstenes reduces the cytotoxic effect of doxorubicin. Since the main mechanism of action of doxorubicin is the stimulation of ROS, androstenes inhibit or block ROS [8]. Androstenes and their derivatives possibly have a cytoprotective effect only in the presence of a stressor or pathology, and this effect is nonspecific in relation to normal and tumor cells.

Another report demonstrated not only a receptor-mediated but also a enzyme-mediated cellular response to DHEA. DHEA inhibited the growth of HeLa and WI-38 cells, and this inhibition could be attenuated by the addition of a combination of four deoxy- and four ribonucleosides to the culture medium [119]. This antiproliferative effect of DHEA may be related to glucose-6-phosphate dehydrogenase antagonism [119]. Indeed, multiple in vivo and in vitro studies conducted on rodents suggest that DHEA prevents carcinogenesis through the inhibition of G6PD or other carcinogen-metabolizing enzymes [26]. Therefore, the expression and activity of enzymes in a particular organ or tissue play an important role in mediating the effects of DHEA.

Regarding the prostate, local enzymatic activity and expression play a role in how DHEA works, either promoting or preventing prostate cancer. The enzymatic microenvironment may underlie immune system-aggravated cancer progression and contribute to prostatic inflammatory atrophy, which is the earliest event of prostate cancer, wherein resident inflammatory cells might induce the metabolism of endogenous DHEA-S and DHEA to DHT, increasing androgenic activity and forming a proliferative microenvironment [26].

### 6.5. Radioprotective Effect of Androstenes

During the metabolism of DHEA, fast (30 min) and slow (several days) metabolites are formed, and since 5-AED is a slow metabolite, at a high dose of DHEA (more than 50 mg/day), it may be effective as a radioprotector and immunostimulant. A dose of 300 mg/day is the maximum dose for humans [12]. Therefore, synthetic esters of steroids can have a prolonged action, and hence a lower dose of esters is needed.

Unlike DHEA, 5-AED is not used as a drug; its clinical studies have not been completed. However, there is convincing evidence that 5-AED is a radioprotector and radiomitigator, protecting the body from radiation exposure [94]. In addition, 5-AED is an immunostimulant [94]. These cytoprotective, radioprotective, and immunomodulatory properties may be useful in radiotherapy for tumor diseases.

There are little data on the effect of 5-AED on inflammation criteria. There are random references to 5-AED normalizing the concentration of IL-6, though 5-AED reduces a high concentration of IL-6 and increases a low concentration if necessary during acute inflammation [120]. As was shown for rat males with traumatic hemorrhages and subsequent sepsis, the decreased IL-2 and IFN-γ synthesis by splenocytes was weakened by the administration of 5-AED. The survival rate of the animals in this model was also improved by 5-AED treatment [120].

In male rats, 5-AED treatment suppresses oxidative stress by increasing the antioxidant enzyme superoxide dismutase (SOD), decreasing the malondialdehyde (MDA) levels, and inhibiting the inflammatory pathway by decreasing toll-like receptor 4 (TLR4), nuclear factor kappa B (NFκB), and high mobility group block 1 (HMGB1) (Figure 3). In addition, 5-AED reduces the production of transforming growth factor beta 1 (TGFβ1), exerting an antifibrosis action, which is highly beneficial in post-operative periods [121].

The cytoprotective effect of DHEA on endothelial cells, shown by Liu et al., is an important mechanism in the fight with the consequences of irradiation and other damages. Hence, not only 5-AED but also DHEA itself may be a quite effective radioprotector and even radiomitigator [14].

### 6.6. Immunostimulatory Action of Androstenes

A series of articles written by Roger M. Loria is devoted to the immunostimulating effects of androstenes.

Reports detail the role of 5-AED and DHEA as immunomodulators which upregulate the immune response to resist viral, bacterial, and parasite infections, including lethal ones. Androstene steroids increase the levels of IL-2, IL-3, and IFNγ cytokines associated with TH1. Analogous to hydrocortisone, DHEA and 5-AED suppress inflammation but do not reduce immunity and help to maintain the TH1/TH2 balance and immune homeostasis [122].

Subcutaneous administration of both DHEA and 5-AED protected mice from a number of various lethal infections, but 5-AED was up to 100 times more effective than DHEA against lethal coxsackievirus B4 (CB4) infection. The protective effect of 5-AED was accompanied by a 3–4 fold increase in the spleen and thymus in animals infected with the virus. DHEA had a similar effect only at a dose exceeding the threshold. However, none of the androstenes showed significant antiviral action in vitro. Also, these steroids had no effect on the titers of the virus in the tissues in vivo. Thus, the mechanisms of the antiviral action of DHEA and 5-AED require further study to create a new type of protection against infectious agents [122].

For 5-AED, there is little information about the molecular mechanisms of its immunostimulatory effect, which seems surprising since 5-AED has a protective action in vivo against infections [122]. However, it is well known that 5-AED acts as a radioprotector by activating the NF-kB signaling pathway [123], which leads to a significant increase in the secretion of G-CSF followed by an increase in the levels of other cytokines, such as IL-6, in the peripheral blood of mice [124]. Further experiments showed that after exposure to ionizing radiation, 5-AED caused increased expression of cytokine mRNA for IL-2, IL-3, IL-6, IL-10, and GM-CSF in the spleen and bone marrow, apparently through the same mechanism as NF-kB activation [125]. At the same time, it is known from other studies that DHEA inhibits the NF-kB signaling pathway [126,127,128]. Thus, 5-AED and DHEA together can be considered a complex regulatory system which modulates the intensity and direction of the immune response (Figure 3).

DHEA and 5-AED in in vitro experiments can develop these effects within a few seconds or minutes, which may indicate that their effect can be realized not only through nuclear receptors (such as the AR, ERa and ERß) but also through some membrane-associated receptors. These rapid non-genomic effects can be mediated through transmembrane receptors such as a transmembrane AR called AR2, G protein-coupled estrogen receptor 1 (GPER1), and zinc transport protein 9 (ZIP9). In contrast to the classical effects of sex steroid hormones carried through nuclear receptors, the physiological significance of rapid non-classical action through membrane-associated and transmembrane steroid receptors is still unclear [129]. Which of the above receptors are most important in the immunostimulating effect of androstenes remains to be studied.

The ability of DHEA and 5-AED to interact with receptors and exhibit immunostimulating activity significantly depends on the location of the hydroxyl group in the C17 position in the tetracyclic androstene molecule. Cytoprotective, immunostimulating, and anti-tumor activity involves only 17β-AED and not 17α-AED, demonstrating a strict relationship between structure and activity [130]. Both DHEA and 17β-AED were named “immunosteroids” by the Loria RM group [131].

### 6.7. Androstenes as Neuroprotectors

DHEA can increase the proliferation of human neural stem cells and positively regulate the number of neurons produced by them. Both NMDA and sigma 1 receptor antagonists (but not GABA receptor antagonists) can completely eliminate the effects of DHEA on stem cell proliferation [132]. These major biological effects of DHEA on the nervous system involve neuroprotection (due to antiapoptotic, antioxidant, anti-inflammatory, and anti-glucocorticoid activities), neurite growth, neurogenesis, and neuronal survival. DHEA also regulates catecholamine synthesis and secretion.

DHEA and DHEA-S predominantly act as noncompetitive antagonists at the GABAA receptor. DHEA-S has more potent antagonistic effects than DHEA [24]. DHEA-S is a positive allosteric modulator of NMDA receptors. According to Manninger, DHEA realizes its neuroprotective action through the following:-Inhibition of NMDA-induced NOS activity (10 µM);-Inhibition of Ca^2+^ influx into the mitochondrial matrix (100 µM);-Inhibition of mitochondrial respiration by acting on complex I of the respiratory chain (100 µM);-Inhibition of NF-κB activation (10–100 µM);-Inhibition of H_2_O_2_ and 4-hydroxynonenal (HNE) production;-Stimulation of SOD, GSH, GSH-peroxidase, and catalase;-Inhibition of lipid oxidation stimulated by H_2_O_2_/FeSO_4_;-A decrease in the serum concentration of TNFα and IL-6 (0.1–100 µM, maximal at 100 µM);-Antagonism to the negative effects of corticosterone on neurogenesis;-Increasing the proliferation of granule cells in the dentate gyrus;-Increasing the number of neural stem cells-Decreasing apoptosis;-Reducing NMDA-induced neurotoxicity (10 µM).

In the brain, DHEA-S modulates the action of the GABAA receptor, the NMDA receptor, and the sigma subtype 1 (σ1) receptor [24]. The neuroprotective and neurogenic effects of DHEA-S and DHEA are realized predominantly through GABA and NMDA receptors [133]. DHEA as well as DHEA-S directly bind to and activate the TrkA, TrkB, TrkC, and p75NTR with high affinity (roughly 5 nM) and are considered important endogenous neurotrophic factors. These findings may explain age-related neurodegenerative diseases when the level of DHEA is low [29,54].

A large number of clinical studies have been and are being conducted, and those concerned with androstene derivatives are among them. For the treatment of various neurological diseases, DHEA derivatives with additional anti-inflammatory and neuroprotective properties which improve cognitive functions were synthesized, such as BNN27 [134].

In the treatment of schizophrenia, DHEA was not effective enough [135]. It only improved the functions of attention, as well as visual and motor skills, and did not affect the clinical psychopathological symptoms of the disease [136].

Future prospects for DHEA and 5-AED involve the development of different regimens and doses to treat Alzheimer’s disease, schizophrenia, multiple sclerosis, post-traumatic stress disorders (PTSD), and Parkinson’s disease [137,138].

## 7. Conclusions

The purpose of this review was to attract more attention to the future clinical application of DHEA and 5-AED. Currently DHEA is used only as a vaginal suppository. Prasterone (or DHEA) is also applied in a wide range of food supplements in different dosage forms. The DHEA metabolite 5-AED (Neumune) as a radioprotector has passed clinical trials, but the final results have not been published.

An important problem is the low bioavailability of both steroids. The promise and new clinical applications of DHEA and 5-AED are presumed to involve the development of dosage forms that offer benefits, such as liposomes and nanoemulsions.

Due to the inherent capacity of the endogenous steroid DHEA to activate and inactivate steroid receptors in peripheral tissues, known as “intracrinology”, the local topical administration of DHEA could be beneficial in the treatment of mucosa atrophy, skin disorders, muscle atrophy, and other diseases. The oral forms of 5-AED and its synthetic derivatives could be highly effective cytoprotectors and immunostimulants.

## Figures and Tables

**Figure 1 pharmaceuticals-17-01186-f001:**
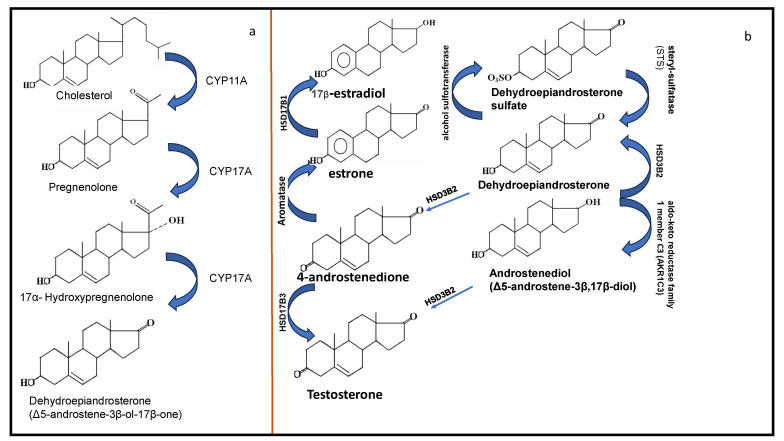
(**a**) Biosynthesis of DHEA. (**b**) Transformation of DHEA. Note: HSD17B3 = hydroxysteroid 17-beta dehydrogenase 3; HSD17B1 = hydroxysteroid 17-beta dehydrogenase 1; HSD3B2 = hydroxy-delta-5-steroid dehydrogenase.

**Figure 2 pharmaceuticals-17-01186-f002:**
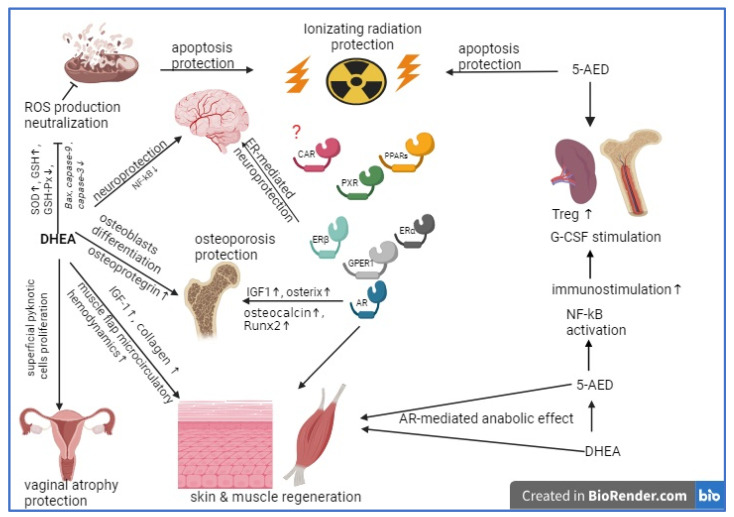
Summary of the pharmacological activities of DHEA and 5-AED with possible clinical applications. Note: DHEA has a direct regenerating action toward vaginal mucosa, skin, muscles, and bones through IGF-1 induction and AR- and ER-mediated pathways. DHEA on different tissues exerts a cytoprotective effect: ROS production neutralization, apoptosis inhibition, and NF-kB inhibition. Meanwhile, 5-AED has a strong immunostimulating effect and a radioprotective effect through Treg proliferation in the spleen and G-CSF production induction. The role of the CAR, GPER, PXR and PPAR in these activities should be identified in future research (noted with question mark).

**Figure 3 pharmaceuticals-17-01186-f003:**
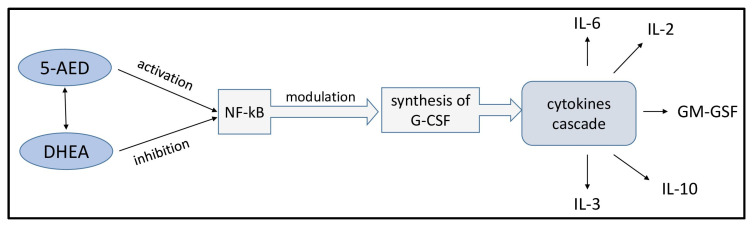
Molecular mechanisms of the immunomodulatory action of 5-AED and DHEA.

**Table 1 pharmaceuticals-17-01186-t001:** DHEA targets according to Drugbank database *.

Target	Action	Organism	Reference
Estrogen receptor alpha (ERα)	Binder (modulator)	Rat	[33]
Estrogen receptor alpha (ERα)	Binder (modulator)	Human	[16]
Estrogen receptor beta (ERβ)	Activator	Humans	[34]
G protein-coupled estrogen receptor (GPER)	Activator	Humans	[35]
γ-aminobutyric-acid-type A receptor (GABA (A))	Antagonist	Humans	[36,37]
N-methyl-d-aspartate (NMDA) receptor	Agonist	Humans	[37]
N-methyl-d-aspartate (NMDA) receptor	Agonist	Rat	[38]
Androgen receptor (AR)	Agonist	Humans	[39,40]
Peroxisome proliferator-activated receptor alpha (PPARα)	Activator	Humans	[34,41]
Sigma non-opioid intracellular receptor 1	Agonist	Humans	[42]
Nuclear receptor subfamily 1 group 1 member 2 (Pregnane X receptor)	Activator	Humans	[43]
Nuclear receptor subfamily 1 group 1 member 2 (Constitutive andronstane receptor (CAR)	Activator	Humans	[44]
GSK-3β glycogen-synthase-kinase-3β (GSK-3β)	Activator	Humans	[45]
Glucose-6-phosphate dehydrogenase (G6PD)	Inhibitor	Humans	[46]

* Table taken from the DrugBank database and modified and supplemented with references.

**Table 2 pharmaceuticals-17-01186-t002:** Summary of constants of binding of hormones (Kis) to steroid receptors.

Name of Steroid	AR	ERα	ERβ	Reference
DHEA	1177 (breast cancer cells) 1259 (prostate cancer cells)	1053	514	[16]
DHEA	-	245	163	[47]
5-AED	210	49	10	[48]
5-AED	-	3.6	0.9	[47]
5-AED	-	100	5	[49]
DHT	0.5 (breast cancer cells) 0.52 (prostate cancer cells)	>5000	1687	[16]
DHT	-	221	73	[47]
DHT	15	-	-	[48]
Testosterone	0.5 (breast cancer cells) 0.52 (prostate cancer cells)	>5000	>500	[16]
E_2_	3.64	0.04	0.1	[16]
E_2_	-	8	7	[48]
E_2_	-	0.12	0.13	[47]

**Table 3 pharmaceuticals-17-01186-t003:** Efficacy summary for primary 12 week trials 1 and 2, with difference from placebo.

Clinical Parameter (Endpoint)	Trial 1: Intrarosa N = 81, Placebo N = 77	Trial 2: Intrarosa N = 157, Placebo N = 325
Dyspareunia Week 12 Mean Severity	−0.40	−0.35
% Superficial Cells Week 12 Mean	4.71	8.46
% Parabasal Cells Week 12 Mean	−45.77	−29.53
Vaginal pH Week 12 Mean	−0.83	−0.67

**Table 4 pharmaceuticals-17-01186-t004:** DHEA, 5-AED, and their derivatives which are being subjected or were subjected to clinical trials and their derivatives.

	Androstene Chemical Name	Androstene Trade Name	Pharmacological Indication	Status	Reference or Link
1	5-androstenediol	Tetragold, Neumune	Radioprotective agent	Investigational	[73,74,75]
2	DHEA	Prasterone	The treatment of dyspareunia associated with menopausal vulvar and vaginal atrophy	FDA approved	[59,60,61,62,63,64,65]
3	3α-ethynyl-androst-5-ene-3β,7β,17β-triol	Triplex (ne3107)	Anti-inflammatory action: collagen-induced arthritis, Alzheimer’s disease, and Parkinson’s disease	Investigational	[76,77]
4	17α,20R-epoxypregn-5-ene-3β,21-diol	BNN27	Neuroprotector	Investigational	[78,79,80]
5	3β-dehydroxy-16α-fluoro-DHEA	Fluasterone	Anti-inflammatory effects in preclinical models of chronic autoimmune diseases (psoriasis, asthma, rheumatoid arthritis, multiple sclerosis, and lupus erythematosus) and lastly for the control of hyperglycemia in adults with Cushing syndrome	Investigational	[81,82,83,84,85]

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
