# Peer review of "Dehydroepiandrosterone and Its Metabolite 5-Androstenediol: New Therapeutic Targets and Possibilities for Clinical Application"

_pharmaceuticals, 2024, doi:10.3390/ph17091186_

Round 1

Reviewer 1 Report

Comments and Suggestions for Authors

In this study authors presented a review for the therapeutic target potential of dehydroepiandrosterone and its metabolite 5-androstenediol for possible therapeutic application. Overall, the manuscript is concise and written well. Some basic points need to be addressed before its consideration in Pharmaceuticals.

1.      Kindly change the title more attractive and omit the term “Androstenes” as it looking little bit create confusion to readers.

2.        Abstract: Abbreviation AR, ER, PXR, LXR is mentioned without its full term. Its better to avoid abbreviations in abstract section.

3.      Write full term of DHEA and 5-AED in introduction section first time as it cannot continue with abstract section.

4.       The available research article relevant to therapeutic target potential of dehydroepiandrosterone need to mention in introduction section

5.      Authors need to provide the schematic presentation of Biosynthesis and transformation of DHEA

6.      The conclusion lacks some basic components. It should be re-written in a well-structured manner and should be concise. It should cover a summary of the problem(s), available relevant information’s and recommendation(s).

Author Response

Dear Reviewer,

Thank you very much for the helpful comments.

Point 1.  Kindly change the title more attractive and omit the term “Androstenes” as it looking little bit create confusion to readers.

We agree with this comment. We have changed the title:

Dehydroepiandrosterone and its metabolite 5-androstenediol: new therapeutic targets and possibilities for clinical application

 Point 2.  Abstract: Abbreviation AR, ER, PXR, LXR is mentioned without its full term. Its better to avoid abbreviations in abstract section.

We agree with this comment. We have changed the Abstract, deleted abbreviations and have added additional section “Abbreviations”.

 Point 3.  Write full term of DHEA and 5-AED in introduction section first time as it cannot continue with abstract section.

We agree with this comment. We have written full term of DHEA and 5-AED in the introduction section first time.

 Point  4.  The available research article relevant to therapeutic target potential of dehydroepiandrosterone need to mention in introduction section

We agree with this comment. We have added  7 references relevant to therapeutic target potential of dehydroepiandrosterone in the introduction section.

Point  5.      Authors need to provide the schematic presentation of Biosynthesis and transformation of DHEA

We agree with this comment. We have added schematic presentation of biosynthesis and transformation of DHEA. Now it is Fig.1.

  1. The conclusion lacks some basic components. It should be re-written in a well-structured manner and should be concise. It should cover a summary of the problem(s), available relevant information’s and recommendation(s).

We agree with this comment. The key sections of the article have been re-written and structured. The recommendations follow from the conclusion.

It was: Currently DHEA is used only as a vaginal suppository. Prasterone (or DHEA) is also applied in a wide range of food supplements in different dosage forms. The DHEA metabolite 5-AED (Neumune) as a radioprotector has passed clinical trials, but the final results have not been published. The DHEA derivative fluasterone, a fluorinated synthetic analogue of DHEA, has been investigated, but the examination was also stopped due to the low oral bioavailability of the compound. Despite this, in May 2024 Fluasterone was presented by the Society for Endocrinology as a drug for the treatment of the metabolic effects of hypercortisolemia and the Cushing’s syndrome.

BNN27, a synthetic neurosteroid and "microneurotrophin" and an analogue of the endogenous neurosteroid DHEA was described as an NGF mimetic and was proposed as a potential novel treatment for neurodegenerative diseases and brain trauma; the studies are going on. The most promising steroid in this line is Triolex (ne3107), as it already has passed clinical trials.

Due to the inherent capacity of the endogenous steroid DHEA to activate and inactivate steroid receptors in peripheral tissues, known as “intracrinology”, the local topical administration of DHEA could be beneficial in the treatment of mucosa atrophy, skin disorders, muscle atrophy, and other diseases.

The promise and new clinical applications of DHEA and 5-AED are presumed to involve the development of derivatives with greater oral bioavailability or the use of dosage forms that offer benefits, such as liposomes and nanoemulsions.

An important problem is the exact identification of targets of pharmacological action of DHEA, in particular, the role of membrane receptors such as mAR, GPER, PGRMC1,2 and mPR in their effects. The way of administration and regimens should be carefully taken into account before the initiation of clinical trials. Due to a very low toxicity of DHEA and its intracrine action, future prospects of the clinical usage of DHEA should include the development of dosage forms for the local external use and the design of DHEA derivatives with improved bioavailability for the oral administration.

It became: Dehydroepiandrosterone and its sulfate are the most abundant steroids in humans. The metabolism of dehydroepiandrosterone can differ significantly depending on the organ or tissue and the subtype of steroid receptors expressed in it. For dehydroepiandrosterone as a precursor of all steroid hormones, intracrine hormonal activity is inherent. This unique feature could be beneficial for the medicinal application, especially for the local treatment of various pathologies. At present, the clinical use of dehydroepiandrosterone is limited by its 6.5 mg vaginal suppositories Intrarosa® (Prasterone) for the treatment of vaginal atrophy and dyspareunia, while the dehydroepiandrosterone synthetic derivatives Triplex, BNN 27, and Fluasterone have the investigational status for the treatment of various diseases. Here we discuss the molecular targets of dehydroepiandrosterone, which open future prospects to expand its indications for use. Dehydroepiandrosterone as an oral drug is surmised to have promise in the treatment of osteoporosis, cachexia, and sarcopenia; and 10% unguent, for the skin and muscle regeneration.

 5-Androstenediol, a metabolite of dehydroepiandrosterone, is a promise candidate for the treatment of acute radiation syndrome and as an immunostimulating agent during radiopharmaceutical therapy. The design and synthesis of new 5-androstenediol derivatives with increased bioavailability may lead to the appearance of highly effective cytoprotectors on the pharmaceutical market. The argumentations for the new clinical applications of these steroids and novel insights in their mechanisms of action are discussed.

Reviewer 2 Report

Comments and Suggestions for Authors

The topic chosen for the present review “Androstenes dehydroepiandrosterone and its metabolite 5-androstenediol: new therapeutic targets and possibilities for clinical application” is interesting; it deals with a wide range of clinical applications of these two steroid hormones: DHEA and its metabolite, 5-AED.

But, the manuscript requires a major revision, because it provides a lot of information about DHEA and 5-AED clinical applications, but without a good systematization of them. It is very difficult to read, to follow, there are many abbreviations, some explained, some not. The entire manuscript contains only one figure (Fig. 2). "Fig. 1" is also mentioned, but it is actually missing.

Abstract -  must be completely rewritten. After the presentation of DHEA and its importance in the human body (1-2 phrases), the purpose of the review should be presented very concisely and what it includes, what will be discussed in it, so that the reader can get an idea from the beginning. In its current form, the Abstract contains far too much, too detailed, unnecessary information related to DHEA targets (lines 13-17). Some ideas should be mentioned about the current clinical applications of these two steroids, and possible applications in the future.

Introduction - first of all, the importance of these two steroids must be highlighted, why was it such a discussed topic in recent years? Then continued with their mode of action, even if it is not fully known or understood, and then with specific targets, etc...

For example, the following sentence is unintelligible (Lines 36-39): “DHEA can serve as a precursor for androstenediol, androstenedione, estrone, testosterone (T), dihydrotestosterone (DHT), and 17β-estradiol (E2) and the biological effects of DHEA can partially intersect with the effects of DHEA metabolites” - What are all these terms listed? It should be explained. What are the “biological effects”? What are “DHEA metabolites” and what effects could they have?

Line 72: “DHEA and its sulfate are the most abundant steroid hormones in the human body. “ - This should be the opening sentence, because the Introduction started with the presentation of the activity of these hormones and not with their actual presentation.

Lines 77-78: “. The main well-established pharmacological effects of DHEA are antiglucocorticoid action” - what does this action consist of? it should be explained, if it is still the "main action" of DHEA

Line 96: Biosynthesis would be much easier to understand if a diagram were made, to better understand where it starts from and what are the stages of this synthesis

4. Current clinical use of DHEA and 5-AED: a very extensive subchapter, with a lot of information about clinical tests. Somehow it must be divided into several sections, either considering the way of administering the doses, or the medical conditions to be treated, for example. It is a collection of mixed information, which refers to both DHEA and 5-AED, nothing is understood anymore.

6. Conclusions: must be rewritten; It should start with the purpose of the review, presenting the importance of the two steroids and then what are the main areas of application, present and in perspective.

I recommend paying close attention to abbreviations, many are not explained.

And I recommend adding more figures. The inclusion of original figures must also be taken into account.

Comments on the Quality of English Language

Moderate editing of English language required; it is quite difficult to read and follow

Author Response

Dear Reviewer,

Thank you very much for the helpful comments.

The topic chosen for the present review “Androstenes dehydroepiandrosterone and its metabolite 5-androstenediol: new therapeutic targets and possibilities for clinical application” is interesting; it deals with a wide range of clinical applications of these two steroid hormones: DHEA and its metabolite, 5-AED.

Point 1.

But, the manuscript requires a major revision, because it provides a lot of information about DHEA and 5-AED clinical applications, but without a good systematization of them. It is very difficult to read, to follow, there are many abbreviations, some explained, some not. The entire manuscript contains only one figure (Fig. 2). "Fig. 1" is also mentioned, but it is actually missing.

Abstract -  must be completely rewritten. After the presentation of DHEA and its importance in the human body (1-2 phrases), the purpose of the review should be presented very concisely and what it includes, what will be discussed in it, so that the reader can get an idea from the beginning. In its current form, the Abstract contains far too much, too detailed, unnecessary information related to DHEA targets (lines 13-17). Some ideas should be mentioned about the current clinical applications of these two steroids, and possible applications in the future.

Reply 1.

We agree with this comment. The Abstract have been re-written and structured. Line 13-17 deleted. Some ideas are mentioned about the current clinical applications of these two steroids, and possible applications in the future.

It was: Currently DHEA is used only as a vaginal suppository. Prasterone (or DHEA) is also applied in a wide range of food supplements in different dosage forms. The DHEA metabolite 5-AED (Neumune) as a radioprotector has passed clinical trials, but the final results have not been published. The DHEA derivative fluasterone, a fluorinated synthetic analogue of DHEA, has been investigated, but the examination was also stopped due to the low oral bioavailability of the compound. Despite this, in May 2024 Fluasterone was presented by the Society for Endocrinology as a drug for the treatment of the metabolic effects of hypercortisolemia and the Cushing’s syndrome.

BNN27, a synthetic neurosteroid and "microneurotrophin" and an analogue of the endogenous neurosteroid DHEA was described as an NGF mimetic and was proposed as a potential novel treatment for neurodegenerative diseases and brain trauma; the studies are going on. The most promising steroid in this line is Triolex (ne3107), as it already has passed clinical trials.

Due to the inherent capacity of the endogenous steroid DHEA to activate and inactivate steroid receptors in peripheral tissues, known as “intracrinology”, the local topical administration of DHEA could be beneficial in the treatment of mucosa atrophy, skin disorders, muscle atrophy, and other diseases.

The promise and new clinical applications of DHEA and 5-AED are presumed to involve the development of derivatives with greater oral bioavailability or the use of dosage forms that offer benefits, such as liposomes and nanoemulsions.

An important problem is the exact identification of targets of pharmacological action of DHEA, in particular, the role of membrane receptors such as mAR, GPER, PGRMC1,2 and mPR in their effects. The way of administration and regimens should be carefully taken into account before the initiation of clinical trials. Due to a very low toxicity of DHEA and its intracrine action, future prospects of the clinical usage of DHEA should include the development of dosage forms for the local external use and the design of DHEA derivatives with improved bioavailability for the oral administration.

It became: Dehydroepiandrosterone and its sulfate are the most abundant steroids in humans. The metabolism of dehydroepiandrosterone can differ significantly depending on the organ or tissue and the subtype of steroid receptors expressed in it. For dehydroepiandrosterone as a precursor of all steroid hormones, intracrine hormonal activity is inherent. This unique feature could be beneficial for the medicinal application, especially for the local treatment of various pathologies. At present, the clinical use of dehydroepiandrosterone is limited by its 6.5 mg vaginal suppositories Intrarosa® (Prasterone) for the treatment of vaginal atrophy and dyspareunia, while the dehydroepiandrosterone synthetic derivatives Triplex, BNN 27, and Fluasterone have the investigational status for the treatment of various diseases. Here we discuss the molecular targets of dehydroepiandrosterone, which open future prospects to expand its indications for use. Dehydroepiandrosterone as an oral drug is surmised to have promise in the treatment of osteoporosis, cachexia, and sarcopenia; and 10% unguent, for the skin and muscle regeneration.

 5-Androstenediol, a metabolite of dehydroepiandrosterone, is a promise candidate for the treatment of acute radiation syndrome and as an immunostimulating agent during radiopharmaceutical therapy. The design and synthesis of new 5-androstenediol derivatives with increased bioavailability may lead to the appearance of highly effective cytoprotectors on the pharmaceutical market. The argumentations for the new clinical applications of these steroids and novel insights in their mechanisms of action are discussed.

Point 2.

Introduction - first of all, the importance of these two steroids must be highlighted, why was it such a discussed topic in recent years? Then continued with their mode of action, even if it is not fully known or understood, and then with specific targets, etc...

For example, the following sentence is unintelligible (Lines 36-39): “DHEA can serve as a precursor for androstenediol, androstenedione, estrone, testosterone (T), dihydrotestosterone (DHT), and 17β-estradiol (E2) and the biological effects of DHEA can partially intersect with the effects of DHEA metabolites” - What are all these terms listed? It should be explained. What are the “biological effects”? What are “DHEA metabolites” and what effects could they have?

Line 72: “DHEA and its sulfate are the most abundant steroid hormones in the human body. “ - This should be the opening sentence, because the Introduction started with the presentation of the activity of these hormones and not with their actual presentation.

Lines 77-78: “. The main well-established pharmacological effects of DHEA are antiglucocorticoid action” - what does this action consist of? it should be explained, if it is still the "main action" of DHEA

Line 96: Biosynthesis would be much easier to understand if a diagram were made, to better understand where it starts from and what are the stages of this synthesis

Reply 2.

We agree with this comment. The Introduction have been re-written and structured.

  • We have changed “DHEA can serve as a precursor for androstenediol, androstenedione, estrone, testosterone (T), dihydrotestosterone (DHT), and 17β-estradiol (E2) and the biological effects of DHEA can partially intersect with the effects of DHEA metabolites”

To  “DHEA serves as a precursor for the steroids androstenediol, androstenedione, testosterone (T), dihydrotestosterone (DHT), estrone and 17β-estradiol (E2) and the biological effects of DHEA can partially intersect with the androgenic and estrogenic activities of these steroids”

  • “DHEA and its sulfate are the most abundant steroid hormones in the human body. “ is now the opening sentence
  • We have changed and added references “. The main well-established pharmacological effects of DHEA are antiglucocorticoid action” to

“One of the well-established pharmacological effects of DHEA is the antiglucocorticoid action [17], protecting mice against dexamethasone-induced atrophy of the spleen and the thymus [18]. The exact mechanism of the anti-glucocorticoid effect of DHEA is not known. DHEA does not bind to the nuclear glucocorticoid receptors [18]. The down regulation of mRNA of glucocorticoid receptors by DHEA may be the reason of the observed anti-glucocorticoid effects of DHEA [19].”

  • Line 96: Biosynthesis would be much easier to understand if a diagram were made, to better understand where it starts from and what are the stages of this synthesis.

We have added schematic presentation of biosynthesis and transformation of DHEA. Now it is Fig.1.

Point 3.

  1. Current clinical use of DHEA and 5-AED: a very extensive subchapter, with a lot of information about clinical tests. Somehow it must be divided into several sections, either considering the way of administering the doses, or the medical conditions to be treated, for example. It is a collection of mixed information, which refers to both DHEA and 5-AED, nothing is understood anymore.

Reply 3.

We agree with this comment.

We have deleted extensive information about clinical tests and devided into 2 sections: “DHEA vaginal gel” and “Oral DHEA”

Point 4.

  1. Conclusions: must be rewritten; It should start with the purpose of the review, presenting the importance of the two steroids and then what are the main areas of application, present and in perspective.

Reply 4.

We agree with this comment.

The key sections of the article have been re-written and structured. The conclusion section starts with the purpose of the review, presenting the importance of the two steroids and then what are the main areas of application, present and in perspective.

It was: Currently DHEA is used only as a vaginal suppository. Prasterone (or DHEA) is also applied in a wide range of food supplements in different dosage forms. The DHEA metabolite 5-AED (Neumune) as a radioprotector has passed clinical trials, but the final results have not been published. The DHEA derivative fluasterone, a fluorinated synthetic analogue of DHEA, has been investigated, but the examination was also stopped due to the low oral bioavailability of the compound. Despite this, in May 2024 Fluasterone was presented by the Society for Endocrinology as a drug for the treatment of the metabolic effects of hypercortisolemia and the Cushing’s syndrome.

BNN27, a synthetic neurosteroid and "microneurotrophin" and an analogue of the endogenous neurosteroid DHEA was described as an NGF mimetic and was proposed as a potential novel treatment for neurodegenerative diseases and brain trauma; the studies are going on. The most promising steroid in this line is Triolex (ne3107), as it already has passed clinical trials.

Due to the inherent capacity of the endogenous steroid DHEA to activate and inactivate steroid receptors in peripheral tissues, known as “intracrinology”, the local topical administration of DHEA could be beneficial in the treatment of mucosa atrophy, skin disorders, muscle atrophy, and other diseases.

The promise and new clinical applications of DHEA and 5-AED are presumed to involve the development of derivatives with greater oral bioavailability or the use of dosage forms that offer benefits, such as liposomes and nanoemulsions.

An important problem is the exact identification of targets of pharmacological action of DHEA, in particular, the role of membrane receptors such as mAR, GPER, PGRMC1,2 and mPR in their effects. The way of administration and regimens should be carefully taken into account before the initiation of clinical trials. Due to a very low toxicity of DHEA and its intracrine action, future prospects of the clinical usage of DHEA should include the development of dosage forms for the local external use and the design of DHEA derivatives with improved bioavailability for the oral administration.

It became: The purpose of this review was to attract more attention to the future clinical application of DHEA and 5-AED. Currently DHEA is used only as a vaginal suppository. Prasterone (or DHEA) is also applied in a wide range of food supplements in different dosage forms. The DHEA metabolite 5-AED (Neumune) as a radioprotector has passed clinical trials, but the final results have not been published.

An important problem is the low bioavailability of both steroids. The promise and new clinical applications of DHEA and 5-AED are presumed to involve the development of dosage forms that offer benefits, such as liposomes and nanoemulsions.

    Due to the inherent capacity of the endogenous steroid DHEA to activate and inactivate steroid receptors in peripheral tissues, known as “intracrinology”, the local topical administration of DHEA could be beneficial in the treatment of mucosa atrophy, skin disorders, muscle atrophy, and other diseases. The oral forms of 5-AED and its synthetic derivatives could be highly effective cytoprotectors and immunostimulants.

Point 5.

I recommend paying close attention to abbreviations, many are not explained.

Reply 5.

We agree with this comment. We have added additional section “Abbreviations”.

Point 6.

And I recommend adding more figures. The inclusion of original figures must also be taken into account.

Reply 6.

We agree with this comment. We have added two figures.

Reviewer 3 Report

Comments and Suggestions for Authors

Figure 1 is missing and needs to be included for clarity and completeness. The manuscript contains an excessive number of tables. To enhance readability, please consolidate the tables and utilize three-line tables where appropriate. Additionally, remove any cited papers or clinical trial information from the tables, as this detracts from the professionalism of the presentation. The manuscript also contains too few figures. Consider including additional figures to visually represent key concepts. For example, the immunostimulatory functions of androgens could be effectively illustrated in a figure.

The manuscript lacks sufficient citations in several areas, particularly between lines 3-5 and 61-64. Ensure that all claims are adequately supported by relevant literature.

When citing existing work, it is important to reference representative figures from the cited studies to provide clearer evidence and enhance the credibility of the manuscript.

Comments on the Quality of English Language

need improve, it is suggested to use concise language 

Author Response

Dear Reviewer,

Thank you very much for the helpful comments.

Point.1

Figure 1 is missing and needs to be included for clarity and completeness.

Reply 1.

We agree with this comment. It is our mistake. We have added schematic presentation of biosynthesis and transformation of DHEA. Now it is Fig.1.

Point.2

The manuscript contains an excessive number of tables. To enhance readability, please consolidate the tables and utilize three-line tables where appropriate

Reply 2.

We agree with this comment. We have utilized three-line table and consolidated the tables and figures. 

Point.3

Additionally, remove any cited papers or clinical trial information from the tables, as this detracts from the professionalism of the presentation.

Reply 3.

We agree with this comment. We have removed any cited papers or clinical trial information from the table and gave new references.

Point.4

Additionally, remove any cited papers or clinical trial information from the tables, as this detracts from the professionalism of the presentation.

Reply 4.

We agree with this comment. We have removed any cited papers or clinical trial information from the table and gave the new references.

Point. 5

Consider including additional figures to visually represent key concepts. For example, the immunostimulatory functions of androgens could be effectively illustrated in a figure.

Reply 5.

We agree with this comment. We included two additional figures.  The immunostimulatory functions of DHEA and 5-AED are illustrated in one of them (Fig.3)

Point 6.

The manuscript lacks sufficient citations in several areas, particularly between lines 3-5 and 61-64. Ensure that all claims are adequately supported by relevant literature.

When citing existing work, it is important to reference representative figures from the cited studies to provide clearer evidence and enhance the credibility of the manuscript.

Reply 6.

We agree with this comment. We included 11 additional references and removed links from the table.  We have added  7 references relevant to therapeutic target potential of dehydroepiandrosterone in the introduction section. We tried to reference the main representative figures from the cited studies.

Round 2

Reviewer 2 Report

Comments and Suggestions for Authors

The authors have modified the manuscript according to the reviewer's requirements.

However, it would have been easier to follow the changes, if they had removed everything that needed to be removed, not just cut the text.

Reviewer 3 Report

Comments and Suggestions for Authors

Thank you for the revision

Comments on the Quality of English Language

good